# The Economic Impact of Rectal Cancer: A Population-Based Study in Italy

**DOI:** 10.3390/ijerph18020474

**Published:** 2021-01-08

**Authors:** Anna Gigli, Silvia Francisci, Giulia Capodaglio, Daniela Pierannunzio, Sandra Mallone, Andrea Tavilla, Tania Lopez, Manuel Zorzi, Fabrizio Stracci, Susanna Busco, Walter Mazzucco, Sara Lonardi, Fortunato Bianconi, Antonio Giampiero Russo, Silvia Iacovacci, Diego Serraino, Gianfranco Manneschi, Mario Fusco, Rosanna Cusimano, Massimo Rugge, Stefano Guzzinati

**Affiliations:** 1Institute for Research on Population and Social Policies, National Research Council, 00185 Rome, Italy; anna.gigli@irpps.cnr.it; 2Centro Nazionale per la Prevenzione delle Malattie e la Promozione della Salute, Istituto Superiore di Sanità, 00161 Roma, Italy; daniela.pierannunzio@iss.it (D.P.); sandra.mallone@iss.it (S.M.); andrea.tavilla@iss.it (A.T.); tania.lopez@iss.it (T.L.); 3Servizio Epidemiologico Regionale, Azienda Zero, 35132 Padova, Italy; giulia.capodaglio@azero.veneto.it; 4Registro Tumori Veneto, Azienda Zero, 35132 Padova, Italy; manuel.zorzi@azero.veneto.it (M.Z.); stefano.guzzinati@azero.veneto.it (S.G.); 5Registro Tumori dell’Umbria, Università di Perugia, 06123 Perugia, Italy; fabrizio.stracci@unipg.it (F.S.); fortunato.bianconi@unipg.it (F.B.); 6UOC Programmazione e Controllo di Gestione, Azienda Sanitaria Locale Latina, 04100 Latina, Italy; s.busco@ausl.latina.it; 7Registro Tumori di Palermo, Università of Palermo, 90133 Palermo, Italy; walter.mazzucco@unipa.it; 8Istituto Oncologico Veneto, IRCCS, 35128 Padova, Italy; sara.lonardi@iov.veneto.it; 9Registro Tumori di Milano, Agenzia per la Protezione della Salute di Milano, 20100 Milano, Italy; agrusso@ats-milano.it; 10Registro Tumori di Latina, Dipartimento di Prevenzione Azienda Sanitaria Locale Latina, 04100 Latina, Italy; registrotumori@ausl.latina.it; 11Registro Tumori del Friuli Venezia Giulia, Centro di Riferimento Oncologico, IRCCS, 33081 Aviano, Italy; serrainod@cro.it; 12Registro Tumori della Toscana, Istituto per lo Studio, la Prevenzione e la Rete Oncologica, 50139 Firenze, Italy; g.manneschi@ispo.toscana.it; 13Registro Tumori Azienda Sanitaria Locale Napoli 3 Sud, 80100 Napoli, Italy; mariofusco2@virgilio.it; 14Azienda Sanitaria Locale 6, 90141 Palermo, Italy; rosanna.cusimano.55@gmail.com; 15Registro Tumori del Veneto, Università di Padova, 35122 Padova, Italy; massimo.rugge@unipd.it

**Keywords:** cancer registry, administrative databases, cost analysis, prevalence, real-world data, patterns of care

## Abstract

Costs of cancer care are increasing worldwide, and sustainability of cancer burden is critical. In this study, the economic impact of rectal cancer on the Italian healthcare system, measured as public healthcare expenditure related to investigation and treatment of rectal cancer patients is estimated. A cross-sectional cohort of 9358 rectal cancer patients is linked, on an individual basis, to claims associated to rectal cancer diagnosis and treatments. Costs refer mainly to years 2010–2011 and are estimated by phase of care, as healthcare needs vary along the care pathway: diagnostic procedures are mainly provided in the first year, surveillance procedures are addressed to chronically ill patients, and end-of-life procedures are given in the terminal status. Clinical approaches and corresponding costs are specific by cancer type and vary by phase of care, stage at diagnosis, and age. Surgery is undertaken by the great majority of patients. Thus, hospitalization is the main cost driver. The evidence produced can be used to improve planning and allocation of healthcare resources. In particular, early diagnosis of rectal cancer is a gain in healthcare budget. Policies raising spreading of and adherence to screening plans, above all when addressed to people living in Southern Italy, should be strongly encouraged.

## 1. Introduction

Costs of cancer care are increasing worldwide [1] because the population of cancer survivors is growing, and costs of expensive treatments more recently introduced are rising. Consequently, the economic sustainability of the impact of cancer is a challenge for high-income countries [2], and more so for countries lacking comprehensive social health insurance systems and other types of social safety nets, where cancer can be a major cause of poverty [3,4,5]. The attempt to estimate the economic burden of cancer involves many researchers in many countries and the production of many studies.

At a national aggregate level, direct [6] as well as indirect [7,8] costs were estimated in the European Union, in Canada [9], in the United States [10], in New Zealand [11], and in Finland [12]. Several studies are population-based and use individual-level data linked to administrative databases. This allows the estimation of costs by phase of care, thus taking into account the fact that healthcare needs of patients vary greatly. For example, diagnostic procedures are mainly provided to patients in their first year after diagnosis, surveillance procedures are addressed to chronically ill patients, end-of-life procedures are given to patients in their terminal status, as seen in studies carried out in the US [13,14,15], England [16], and Canada [17]. In Italy, there are various studies based on clinical cohorts [18], or estimating some type of expenditures [19], or single phases of care [20]. 

Colorectal cancer is the third most commonly diagnosed malignancy and the fourth leading cause of cancer death in the world, accounting for about 1.4 million new cases and almost 700,000 deaths in 2012 [21]. In 2018, there were 704,376 newly diagnosed rectal cancer patients, corresponding to 3.9% of all cancers, and 310,394 people died (3.2% of all cancer deaths) [22]. In Italy in 2010 it is estimated that about 2.6 million residents have experienced a cancer in their lives, and in 2020 it is predicted that figures will increase to 3.6 million because of the combination of the effects of the rising of survival and the aging of the population [23]. With 13% of new cancer diagnoses in 2013, colorectal cancer is one of the most frequent cancers in Italy (third in the list of all cancer sites among males and second among females) [24]. One third of new colorectal cases are rectal cancer cases. Statistical indicators of colon and rectal cancer burden are generally considered together, although survivorship and treatments may vary considerably [25,26]. 

In the literature, most studies consider costs in different states of the disease for colon and rectum cancers combined [27,28,29]. However, treatments following a rectal cancer diagnosis are more complex (they may include, for example, radiotherapy, temporary stoma) and require longer hospitalization periods, and this affects costs in the initial phase of care. These findings suggest keeping separate analyses for rectal and colon cancer care. 

In this paper, estimation of direct costs of rectal cancer care in areas covered by population-based cancer registries in Italy is presented. The idea is to identify those costs related to the diagnosis and treatment of rectal cancer. In this study, information supplied by various data sources on an individual basis is used (linking administrative regional-based healthcare sources to cancer registry’s source) in order to build patterns of patient care and individual cost profiles. In estimating the economic burden of the disease over one year, a prevalence approach is adopted. The findings described in this paper derive from the Epicost study [30], the first attempt in Italy to provide population-based estimates of direct cancer costs across the patient pathway.

## 2. Materials

### 2.1. Data Sources

In Italy, a public welfare system guarantees universal healthcare. The National Health Service (NHS) is centrally organized under the Ministry of Health and is administered on a regional basis (19 regions and 2 provinces). Hospitals, clinics and ambulatories authorized by the Ministry, as well as pharmacies (for prescriptions of drugs reimbursed by the NHS) transmit their claims to the regional health authority in order to be reimbursed. These claims are collected in databases containing information at individual level. There are different levels of harmonization of claims among regions, according to the type of healthcare service: drugs prescribed for treatment at home, available in territorial pharmacy, have the same price over the whole Italian territory; hospital admission claims may diverge from the price set by the Ministry of Health; outpatient claims are defined at regional level, and may vary greatly among regions.

Data from 4 databases are considered: Cancer Registry (CR), Outpatient Services (OPS), Drug Prescriptions (DP), Hospital Discharges (HD). CR provides data on cancer patients, the other sources provide data on healthcare services.

Cancer Registries (CRs) collect data on all cancer diagnoses that occur in every person residing in the area covered by cancer registration. The following information for each patient are included: date of birth, date of diagnosis, gender, vital status, site of primary tumor, morphology code, diagnostic confirmation; furthermore, according to the study protocol, other two variables have been provided for most patients: stage at diagnosis, diagnosis modality (patient screened Vs non-screened).

For each type of healthcare service considered in this study (hospitalization, outpatient service, drug prescription), information on each individual is collected and includes, in accordance with the Italian data protection law, an anonymous identifier code able to link each cancer patient in the CR database. 

The HD source contains hospital admissions, each record referring to a single admission and discharge of a single patient. It includes demographic information (date of birth, sex, place of birth, place of residence), clinical information (type of diagnosis, interventions and procedures coded by the International Classification of Diseases, 9th Revision-Clinical Modification (ICD9-CM) [31], date of admission, and date of discharge), administrative information (coded by the DRG coding system), total claim in euros.

The OPS source includes information on outpatient services (such as outpatient interventions, diagnostic tests, etc.), each record referring to a single outpatient episode occurring to a patient. It contains the type of procedure, the date of the episode and total claim in Euros. Each service is coded according to the ICD9-CM coding system. However, each region sets its cost, and decides whether to add more codes corresponding to extra services not included in the ICD9-CM claim list. This is the case of region Lombardia, where a number of services (chemotherapy, immunotherapy, blood tests, oncologist appointment, other specialists’ appointments) provided to a patient in the same day are grouped in a single claim. The claim is not comparable with other regions, since it contains the price of high cost drugs, not included in other regional databases. 

Notice that chemotherapy can be administered either in outpatient or in hospital settings, and related information is included in the Outpatient (OPS) or Hospital discharge (HD) database, respectively.

The DP source includes information on prescription drugs which are sold by pharmacy. Each record refers to a single drug (coded according to the Anatomical Therapeutic Chemical (ATC) classification system [32]), and contains drug code, date of prescription, and total claim in Euros. Drugs may be administered to patients in three settings: hospital, outpatient clinic, pharmacy. The DP database contains detailed information (including molecule and corresponding ATC) only on drugs prescribed to patients and sold by pharmacies. The OPS database includes generic information on chemotherapy drugs administered in outpatient (not molecule nor ATC). The HD database includes the cost of drugs administered during hospital stay in the DRG system, which assigns an overall reimbursement for treatments, procedures, interventions, drugs, and does not contain detailed information (not molecule nor ATC). Finally, high cost drugs, such as biological drugs, monoclonal antibodies, etc. are included in a different database, which was not used in our study, because during data collection we discovered that the information processing and the refund system was widely variable, in terms of completeness, from region to region. In conclusion, detailed information on costs of drugs is available for drugs sold by pharmacies, only.

### 2.2. Study Cohort

This study involves 8 population-based CRs having a minimum of 8 years of cancer registration: Milano, Friuli Venezia Giulia (VG), Veneto in the North; Firenze-Prato, Umbria, Latina in the Centre; Palermo, Napoli in the South; overall, they cover just over 10 million people, corresponding to about one sixth of the Italian population. We use a cross-sectional study design: the study cohort includes patients diagnosed with malignant rectal cancer (ICD9-CM C19, C20) in the most recent 8 years of incidence and still alive at prevalence date (prevalence cohort), as illustrated in Table 1. Each CR uses the most updated data at the time of case extraction. Thus, prevalence date varies among CRs: from 2009 (January 1st) to 2013 (January 1st). In each CR, administrative data used for cost analysis is available for a 24-month period centered around prevalence date. 

Persons who were previously diagnosed of cancer in the five years before diagnosis of rectal cancer, or persons diagnosed with subsequent cancer in the year after diagnosis of rectal cancer, were excluded. Prevalent cases are followed for one year after prevalence date, with respect to their vital status. 

## 3. Methods

### 3.1. Phase of Care Prevalence

Each patient enters in the study for an interval of 12 months, except those who die in less than 12 months after diagnosis (short-term survivors, accounting for 1.3% of the study cohort) and those who are lost within 12 months after prevalence date (cases with censored follow up, 0.12% of the study cohort). 

We define 3 phases of care: initial (12 months after cancer diagnosis); continuing (time elapsed between initial and final); final (last 12 months before death due to cancer). Phases of care are mutually exclusive. Although during her/his life span each patient may span across several phases, on prevalence date each individual is associated to one single phase, depending on the interval between prevalence date and diagnosis date, and on the possible occurrence of death for rectal cancer during the following year. Notice that in case a patient dies for causes other than cancer, his/her follow-up is censored, and the case is assigned to the initial or continuing phase of care. Causes of death are classified according to International Classification of Diseases Tenth Revision (ICD-10). Causes of death other than cancer are S00–T98 (injury, poisoning and certain other consequences of external causes such as burn, frostbite, etc.), V01–Y98 (external causes of morbidity and mortality, such as transport accident, drowning, exposure to forces of nature, etc.).

Figure A1 in Appendix A illustrates the study design and the assignment of each case to the corresponding phase of care.

### 3.2. Definition and Calculation of Costs by Phase of Care

Figure A2 in Appendix A illustrates the periods when data for cost analysis is available for each CR. Each prevalent case is linked to the three databases (OPS, HD, DP) in order to trace all episodes referred to the patient during his/her study period. A deterministic linkage is implemented by means of the anonymous identifier code. Only events related to rectal cancer are considered in a list of correlated events, one for each database. Lists are elaborated by oncologists and clinicians on the basis of clinical guidelines and current practice and comprise procedures and diagnoses classified according to ICD9-CM for OPS and HD databases; drugs classified according to ATC for DP database.

Costs (in Euros) correspond to the amount reimbursed to the healthcare providers (the pharmacies, the ambulatories, and the hospitals) by Regional Health Authorities for the services supplied to a patient with rectal cancer.

The indicators below are computed separately for each healthcare service:

Patient monthly cost *C_jk_^f^*: cost payed for patient *j* (*j* = 1,…, *N*) in month *k* (*k* = 1,…, 12) in phase *f* (*f* = initial, continuing, final). Person months *p_jk_^f^* is a binary indicator than equals 1 if patient *j* is alive in month *k* of phase *f*, and 0 otherwise. Patient monthly average cost *C_k_^f^:* cost payed on average for all patients in month *k* of phase *f*, derived as the ratio between costs payed for patients in month *k* and phase *f* and the corresponding person-months
$$C_{k}^{f}=\frac{\sum_{j=1}^{N}C_{jk}^{f}}{\sum_{j=1}^{N}p_{jk}^{f}}$$

Patient annual average cost *C_A_^f^*: average cost in phase *f* payed for a patient in a year A, which is obtained as the ratio between the sum of patient monthly costs and the sum of person-months, multiplied by 12.

A cost profile is an array *C_k_^f^* of 12 patient monthly average costs in each phase of care *f*. In this study, costs for each patient are considered just for one phase, and a cost profile consists of combining the 36 monthly average costs computed for patients in different phases of care: C_1_^initial^,…, C_12_^initial^, C_1_^continuing^,…, C_12_^continuing^,…, C_1_^final^, C_12_^final^. 

Total annual cost: cost in phase *f* payed for all patients in a 12-month period, which is given by the product of the patient annual average cost in phase *f* and the totality of patients belonging to phase *f*. Within the prevalence cohort, we identify groups of patients that are homogeneous regarding to demographic and clinical features which affect patterns of care: age and stage at diagnosis (for the initial phase, only). Every homogeneous group is a match of stage at diagnosis (I, II, III, IV) and age group (15–49, 50–69, 70–79, 80+). Costs of homogeneous groups are calculated by averaging costs over patients of the same group.

### 3.3. Care Patterns by Phase of Disease

For description and interpretation purposes, the following indicators are calculated (in initial phase only) by stage at diagnosis and age at prevalence: cases undertaking at least one surgical intervention; cases undertaking at least one chemotherapy treatment; cases undertaking at least one radiotherapy treatment; cases undertaking neo-adjuvant chemotherapy and/or neo-adjuvant radiotherapy among cases with surgical intervention in initial phase, expressed as percentages.

### 3.4. Statistics

Chi-square test was applied to compare differences in proportion; Cochran Armitage test for trend was applied to check linearity in trends of proportions. Two-sided *p*-values below 0.05 are evaluated as significant. Software SAS 9.4 was used for the statistical analysis.

## 4. Results

The prevalence cohort includes 9358 subjects, over 57% are males (Table 1). Percentage of patients whose stage at diagnosis is missing (Unstaged) varies widely among CRs: from a minimum of 7% in Veneto CR to a maximum of 33% in Friuli VG CR. On average, over 12 months, a patient with a rectal cancer diagnosis has about 1 hospital admissions, 39 outpatient episodes and less than 2 drugs prescribed outside hospital.

Patients from Napoli CR receive significantly higher rates of hospitalizations (*p* < 0.0001), outpatient services (*p* = 0.0005) and drug prescriptions (*p* < 0.0001). Notice that part of the population of Napoli CR was covered with only 3 years of registration, hence the study cohort is characterized by a higher proportion of newly diagnosed patients, who require more treatments and hospital admissions, and by a lower proportion of intermediate patients.

### 4.1. Overall Costs

Figure 1 shows the dynamic along the disease pathway of the average costs sustained per patient, per month by type of service (hospitalization (a) and outpatient services (b)) in each phase of care computed for all patients in the study, regardless of their distribution by age at prevalence, stage at diagnosis or geographic position. Cost estimates for the pool of CRs refer mainly to years 2010–2011. The X-axis measures the time in each phase of care: I_1_,…, I_12_ indicate the 12 months of the initial phase; C_1_,…, C_12_ the 12 months of the continuing phase; F_1_,…, F_12_ the 12 months of the final phase. The Y-axis measures the monthly average cost per patient.

The main driver of costs is hospitalization, followed by outpatient services. Hospital and ambulatory costs are generally higher in the first few months after diagnosis, when diagnostic and surgical procedures are more frequent, and in the last few months before death, when an intensification of care due to disease progression is needed. Costs due to drug prescriptions are negligible with respect to the other two components along the entire disease pathway and are not shown in the figure.

Figure 2 shows the distribution of total annual costs by type of service in each phase of care (a) and the distribution of prevalent cases (b), for the pool of cancer registries. Cost estimates refer mainly to years 2010–2011.

18% of cases are in initial phase of care, and absorb about 53% of costs (45.6% hospitalization, 7.6% outpatient services, and 0.2% drug costs). Almost 73% of prevalent cases are in continuing phase, and absorb almost 27% of costs (17.3% hospitalization, 9.1% outpatient services, and 0.3% drug costs). Finally, 9% of cases are in final phase and absorb 20% of costs (16.5%, 2.8% and 0.6% of costs due to hospitalization, outpatient services and drug prescriptions, respectively).

As well as by phase of care, the amount of resources varies also by type of service and by age. Table 2 describes the patient annual average costs in each phase of care stratified by age group at prevalence, for the pool of cancer registries. Less than 46% of cases in initial and 41% of cases in continuing phase are in the target age of screening programs (age group 50–69); elderly patients (ages 70 and over) account to 48% in initial phase, 55% in continuing phase and 70% in final phase. Generally, healthcare costs decrease as age at prevalence increases, in all types of services and phases of care.

### 4.2. Costs by Cancer Registry

The distribution of costs varies quite considerably across CRs. Table 3 describes the patient annual average cost according to type of healthcare service and phase of care, for each cancer registry, and for the pool of registries. Prevalent cases are homogeneously distributed across CRs (18%, 73% and 9%, in initial, continuing, and final phases, respectively in the pool of registries), with the exception of Napoli, which is characterized by a significantly higher proportion of cases in initial phase of care (25%) and fewer cases in continuing phase (64%) (*p* = 0.0002).

CRs with highest patient annual average costs (across all phases and all services) are Friuli VG and Napoli; CRs with lowest patient annual average costs are Veneto, Latina and Palermo.

Hospitalization costs in the pool of CRs account for 85% and 83% of total costs in initial and final phases, respectively and vary from about 9300 Euros per patient in Milano to nearly 15,000 Euros per patient in Umbria in initial phase, and from about 5000 Euros in Veneto to 12,700 in Friuli VG in final phase.

Outpatient services costs are highest in Veneto, Milano and Friuli VG, and lowest in Firenze (initial and continuing phases) and Latina (final phase). The relative distribution between services varies greatly: in initial phase Firenze accounts for 94% of costs due to hospitalization and 6% to outpatient services, while Veneto accounts for 75% of costs due to hospitalization and 25% to outpatient services; in final phase Firenze accounts for 90% of costs due to hospitalization and 7% to outpatient services, while Veneto accounts for 69% of costs due to hospitalization and 28% to outpatient services. In Veneto there is a tendency to treat patients more frequently in outpatient care, which is less costly than hospitalization.

### 4.3. Focus on Initial Phase of Care

Initial phase accounts for more than half of the total costs. Table 4 illustrates the distribution of cases and average annual costs per patient by stage (which is a proxy for the severity of the disease), type of service and CR. Stage at diagnosis is predictive of treatment and consequently of expenditure: more advanced stages require more expensive treatments. In the pool of CRs, out of 1603 patients in initial phase, 20% are in stage I, 27% in stage II, 26% in stage III, 9% in stage IV. Stage is missing in 18% of cases. Among registries, stages I and II are more frequent than stages III and IV, apart from Milano and Palermo, where patients in early and late stages are almost equivalent. Costs show a stage at diagnosis trend: patients in more advanced stages cost more: cases in stages III and IV cost 50% more than cases in stages I and II (treatments for cases in stage III or IV cost around 18,000 Euros vs. about 12,000 Euros for treating patients in stage I or II). This trend is observed across all CRs in almost all cost components. Total costs vary by stage from a minimum of about 7000 Euros in Palermo stage I to a maximum of over 26,000 Euros in Friuli VG stage IV.

Table 5 focuses on several treatments delivered in initial phase, stratified by stage and age class. Surgery is the most common treatment, and it is received by the great majority of patients, more so in stages II or III. In contrast, chemotherapy is more frequently administered in late stages, particularly for stage IV tumors, radiotherapy more often in stages II and III. There is an inverse trend of treatment by age, with younger patients receiving treatments more often than older patients, the only exception being stage I patients aged 80 and over with respect to radiotherapy and neo-adjuvant radiotherapy A portion of patients with locally advanced rectal cancer who undergo surgery receive neo-adjuvant treatment, that is chemo- and/or radiotherapy within 3 months before surgery, generally more frequently in patients below 70 years of age.

## 5. Discussion

Rectal prevalent cases represent 30% of colon and rectum cancers cases combined, and 34% of costs (data not shown from the Epicost study). Results show that average costs per patient have a U-shape: costs are higher in the first 2–3 months, when diagnostic tests and major surgeries are supplied, as well as in the end-of-life, when palliative care is supplied. Similar results are found elsewhere [16,33,34]. In the initial phase of care, hospitalization costs are highest in the first two months after diagnosis. Outpatient services costs are lower in the first month after diagnosis and then increase up to a maximum in the third month. Such a trend is coherent with the process of care: diagnostic tests and surgery are performed in hospital followed by chemotherapy and/or radiotherapy in an outpatient setting. A similar dual pattern is observed in the end-of-life phase of care: outpatient costs rise up in the first part and drop down in the last month before death, when hospitalization costs rise up.

Hospitalization represents the main cost item (79% of total expenditure), followed by outpatient services (20%) and drug prescriptions (only 1%). Notice that chemotherapy is included among the hospital or outpatient costs, according to the delivering setting. However, the recent increase of costs for antitumor drugs (not considered in this study, but negligible at the time of data collection) could deeply modify the observed pattern, making drug prescriptions costs higher.

Stage at diagnosis greatly influences costs of the initial phase of care, and cases diagnosed with advanced disease absorb 47% resources more than cases diagnosed with early disease. This result is confirmed in a previous study [19].

Age is another determinant of costs, since clinical approaches vary by age: more aggressive (and more expensive) treatments are better tolerated by younger patients, who have higher life expectancy when faced with aggressive treatments, in comparison with older patients, who generally have more co-morbidities.

Some strong points and weaknesses of this study derive from the methodology, others from the available data. Among the strong points:

Ours is a real-world study, i.e., findings are at population level, and there is no selection concerning prognosis or regarding any patient’s demographic and clinical feature. We considered all data sources on claims accessible at the time of the study. We adopted a cross-sectional approach, because it produces more up-to-date results than those obtained with a longitudinal approach. Eight years of follow-up are a time interval long enough to observe the entire pattern of care, provided that in Italy a recent estimate of time to cure for colorectal cancer patients is eight years [35]. Finally, with the phase-of-care framework all clinically significant phases of the disease are considered.

Some weaknesses should be considered, as they may affect the results:

Some data sources are not considered in this study: home care services, nursing facilities for elderly people, emergency room (ER) services, hospices for terminal patients. As a consequence, total costs might be underestimated, depending on the patient features and the phase of the disease: hospices are supplied to end-of-life patients, nursing facilities are usually supplied to elderly cases, home care to either case; ER services are not particularly used in a chronic disease like cancer; further, at the time of data collection, the use of hospice for terminal patients was not routinely implemented and most patients died in hospital. 

In-hospital drugs database is not considered in the analysis, because the archives were incomplete and of poor quality; as a consequence, highly expensive drugs not included in the DRG reimbursement protocol are not taken into consideration and this may lead to an underestimation of pharmaceutical costs, even though at the time of data collection these drugs were scarcely administered in Italy.

Information on stage at diagnosis is not complete, ranging from 63% in Friuli VG CR to 93% in Veneto CR. This is partly due to inefficiencies in some regional healthcare information systems, and partly to migration of patients between regions: when a patient undertakes treatments outside her/his region, a few clinical data (for example, stage) might be missing. This incompleteness might limit the comparability of initial phase costs between cancer registries. 

The surveillance phase includes a combination of cases with varying clinical features and care-patterns: some patients are fully recovered; some others experience relapses; other patients live in chronic conditions. Currently, information collected by CRs does not allow to distinguish these groups of patients.

We presented results by region because in Italy the reimbursement system is regionally based. However, regional comparison of total costs by phase of care is limited by the following confounders: in the outpatient setting, different regions may pay different reimbursement for the same healthcare service, and each region may decide to add extra procedures. Moreover, the same procedure may be supplied by various regions in various settings: chemotherapy, for instance, is given more frequently in outpatient clinic in the Northern regions and in hospital in the Centre-Southern ones. Notwithstanding these limitations and bearing in mind that this study is not focused on comparisons between regions, some arguments regarding how costs of rectal cancer care vary geographically can help in the identification of good practices and best models of healthcare planning. For example, in Veneto several healthcare treatments are shifted from hospital to outpatient setting; in particular, chemotherapy has been administered in outpatient setting since 2007, and this different organization yields lower overall costs.

## 6. Conclusions

To our knowledge, this paper is the first study to estimate, at a population level using micro-data, the economic burden of rectal cancer on the public health system in Italy. Estimation is based on a three-phase pattern of care that considers the whole process of the disease from initial diagnosis to cure/death. Information at individual level comes from various healthcare and administrative databases.

The approach of this study allows policy makers to identify areas with different needs—among healthcare services, among phases of care, and among some patients’ characteristics, such as age and stage. Our model may support policy makers in predicting near-future cancer burden on the basis of different scenarios induced by specific interventions. For example, this study shows that early diagnosis of rectal cancer is a gain in the healthcare budget. Therefore, policies raising the spreading of and adherence to screening plans, above all when addressed to people living in the South of Italy, should be strongly encouraged. Presently, the diffusion and adherence of organized screening programs for colorectal cancer in Italy is very variable among regions [36].

Standardization and completeness of in-hospital drug databases have improved in more recent years. In a future perspective, specific data check procedures developed in the Epicost study will be used to include in-hospital drug database in cost analysis. Furthermore, in the continuing phase groups of patients that are homogeneous in terms of similar care needs will be identified through specific procedures [37].

The type of analysis proposed here can be extended to other countries with diverse healthcare managements and systems, as long as data on healthcare services and related costs at individual level are accessible. As an example, in the ongoing Innovative Partnership for Action Against Cancer (iPAAC) financed by the European Commission, the methodology has been proposed for application to other European countries, such as Belgium, Spain, Norway, and Poland [38].

## Figures and Tables

**Figure 1 ijerph-18-00474-f001:**
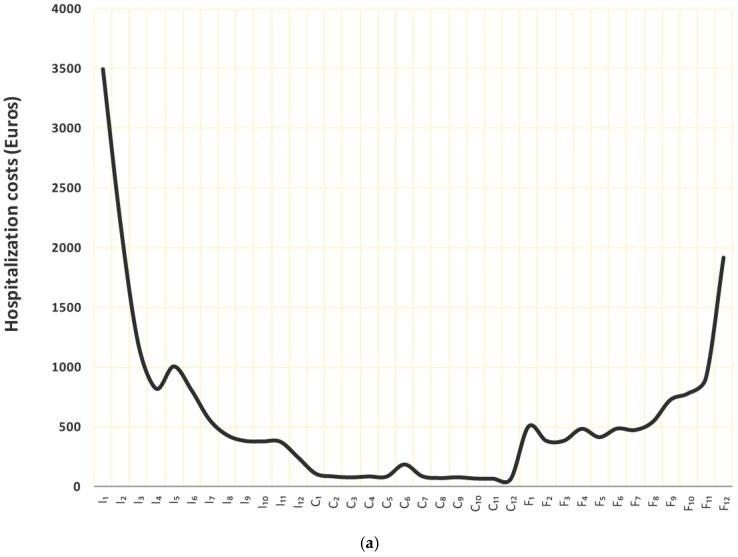
(**a**) Cost profile (or patient monthly average costs) due to hospitalization. Pool of CRs. (**b**) Cost profile (or patient monthly average costs) due to outpatient services. Pool of CRs.

**Figure 2 ijerph-18-00474-f002:**
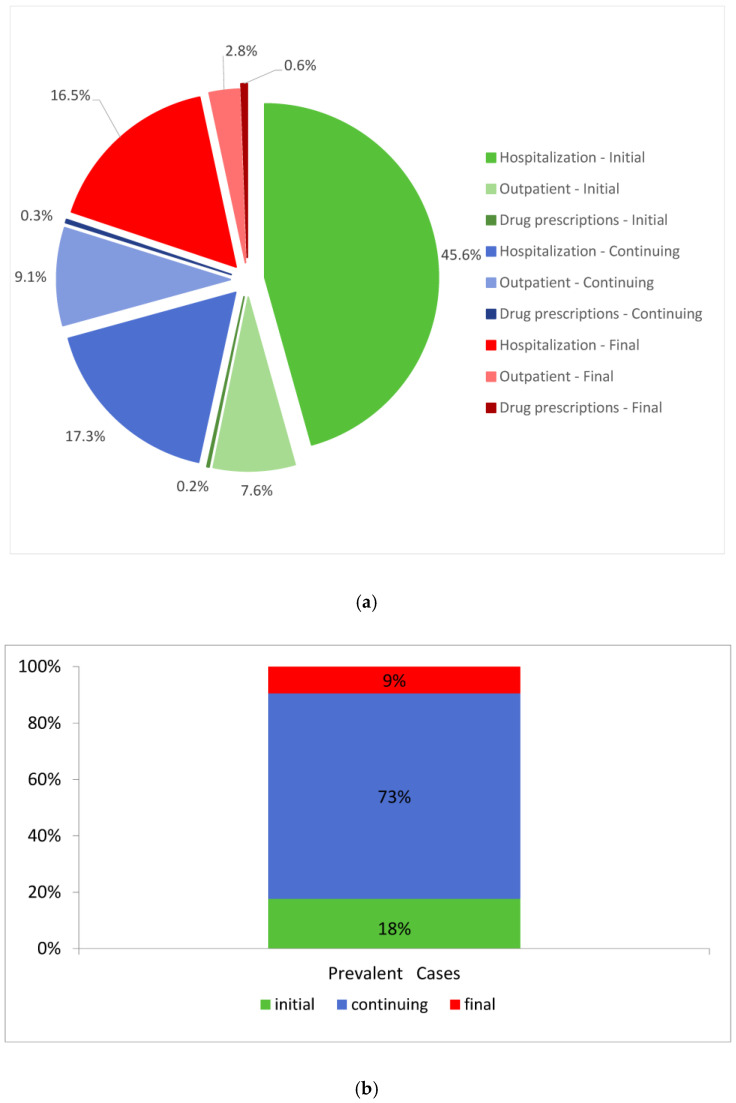
Distribution of total annual costs (**a**) and prevalent cases (**b**) by type of service and phase of care. Pool of CRs.

**Table 1 ijerph-18-00474-t001:** Population coverage, Prevalence, Average number of events in a year by Cancer Registry (CR) and in the pool of CRs.

		Cancer Registry								
		Firenze	Friuli VG	Latina	Milano	Napoli	Palermo	Umbria	Veneto	PooL
Population coverage	Counts	1,211,074	1,219,493	537,590	3,300,881	1,163,644	1,240,830	879,993	629,993	10,183,498
	% Males	47.9	48.3	48.8	48.1	48.5	48.2	48.1	49.3	48.3
	Regional Coverage	33.3%	100%	9.8%	33.7%	20.2%	24.8%	100%	13.0%	27.8%
Prevalence	Date (January 1st)	2009	2010	2011	2013	2011	2011	2011	2010	
	8-year Incidence	2001–2008	2002–2009	2003–2010	2005–2012	2003–2010	2003–2010	2003–2010	2002–2009	
	Cases	1495	1407	477	2671	540	1006	1166	596	9358
	% Males	56.5	57.2	59.7	57.9	52.8	55.9	59	58.2	57.3
	Cases within 1 year	227	270	78	409	128	176	211	105	1604
	Unstaged	21.6%	33.0%	25.4%	11.3%	10.9%	24.4%	11.4%	6.7%	
Average events in a year	Hospital Admissions	0.7	0.7	0.7	0.6	2.1	0.9	0.7	0.5	0.8
	Outpatient Services	35.3	43.8	45	38.8	51.5	38.1	35.2	32.1	39.1
	Drug Prescriptions	1.7	1.7	1.8	1.1	2.5	1.8	2	1.6	1.6

**Table 2 ijerph-18-00474-t002:** Prevalent cases by age at prevalence and patient annual average costs by age at prevalence, phase of care and type of service. Pool of CRs.

	**Initial Phase**				
**Age**	**Prevalent Cases ^a^**	**Hospitalization ^b^**	**Outpatient ^b^**	**Drug Prescription ^b^**	**Total Costs ^c^**
15–49	99	14,425	2787	67	17,279
50–69	739	12,379	2590	58	15,027
70–79	529	12,209	2169	88	14,466
80+	255	10,372	966	27	11,365
	**Continuing Phase**				
**Age**	**Prevalent Cases ^a^**	**Hospitalization ^b^**	**Outpatient ^b^**	**Drug Prescription ^b^**	**Total Costs ^c^**
15–49	281	1446	857	11	2314
50–69	2714	1301	731	21	2053
70–79	2239	912	594	19	1525
80+	1443	641	348	22	1011
	**Final Phase**				
**Age**	**Prevalent Cases ^a^**	**Hospitalization ^b^**	**Outpatient ^b^**	**Drug Prescription ^b^**	**Total Costs ^c^**
15–49	30	16,180	2398	989	19,567
50–69	227	12,497	2786	597	15,880
70–79	233	9598	1621	223	11,442
80+	378	3933	517	112	4562

^a^ Person-years, ^b^ Patient annual average costs in Euros, ^c^ Patient annual average costs in Euros all services combined.

**Table 3 ijerph-18-00474-t003:** Prevalent cases by CR and phase of care and patient annual average costs by CR, type of service and phase of care.

		Cancer Registry								
Phase of Care		Firenze	Friuli VG	Latina	Milano	Napoli	Palermo	Umbria	Veneto	PooL ^a^
Initial	Prevalent Cases ^b^	229	274	79	413	130	178	213	106	1622
	Hospitalization ^c^	13,418	14,353	10,854	9311	13,345	11,362	14,987	9638	12,159
	Outpatient ^c^	841	2807	1686	3509	1785	1041	1349	3150	2021
	Drug Prescription ^c^	13	27	80	154	57	46	24	23	53
	Total costs ^c^	14,271	17,187	12,620	12,973	15,186	12,448	16,360	12,810	14,232
Continuing	Prevalent Cases ^b^	1075	983	350	1989	337	706	812	425	6677
	Hospitalization ^c^	1195	1164	1133	754	2059	777	1394	480	1120
	Outpatient ^c^	398	733	550	745	638	495	465	687	589
	Drug Prescription ^c^	20	21	18	22	21	17	15	23	20
	Total costs ^c^	1612	1919	1701	1522	2717	1289	1874	1190	1728
Final	Prevalent Cases ^b^	168	118	40	219	56	98	119	52	868
	Hospitalization ^c^	7595	12,680	6730	6236	11,231	7494	8592	5143	8213
	Outpatient ^c^	627	2148	608	2365	1640	861	805	2059	1389
	Drug Prescription ^c^	261	342	186	353	348	401	180	211	285
	Total costs ^c^	8483	15,170	7524	8954	13,219	8756	9578	7412	9887
Grand Total costs ^d^		6,423,658	8,385,964	1,886,562	10,345,333	3,630,197	3,975,765	6,145,767	2,252,122	43,206,310

^a^ Pool of Cancer Registries, ^b^ Person-years, ^c^ Patient annual average costs in Euros by type of service and all services combined (Total), ^d^ Grand Total costs for all patients and all types of services.

**Table 4 ijerph-18-00474-t004:** Prevalent cases in initial phase by stage at diagnosis and patient annual average costs in initial phase by stage at diagnosis and type of service.

		Cancer Registry								
	Stage	Firenze	Friuli VG	Latina	Milano	Napoli	Palermo	Umbria	Veneto	Pool ^a^
Prevalent cases ^b^	I	30	80	7	46	14	11	87	44	319
	II	83	36	26	136	50	57	35	16	439
	III	49	51	19	149	33	45	45	27	418
	IV	16	14	6	32	17	20	20	11	136
	NA	49	89	20	45	14	43	24	7	291
Hospitalization ^c^	I	10,912	9086	13,204	7393	9062	6616	11,584	8141	9500
	II	13,543	15,806	8413	10,284	12,027	9798	16,996	8900	11,971
	III	17,528	15,176	14,086	9130	13,545	12,171	18,602	12,367	14,076
	IV	20,752	21,171	14,005	8892	17,547	16,416	18,381	11,207	16,046
	NA	6624	15,728	8506	5327	12,610	10,606	11,584	5241	9528
Outpatient ^c^	I	790	951	2896	993	972	540	1013	876	1129
	II	781	2769	1267	3357	1847	731	1236	3392	1922
	III	969	5081	1789	4146	1967	1269	1845	5693	2845
	IV	1224	5178	3504	4936	1904	1810	1437	5696	3211
	NA	701	2872	1233	3220	1955	1003	1722	3080	1973
Drug Prescription ^c^	I	8	8	35	14	27	6	13	5	15
	II	15	13	16	81	20	8	55	47	32
	III	4	26	204	264	89	66	17	24	87
	IV	21	32	168	246	123	141	34	6	96
	NA	14	49	36	47	23	42	17	105	42
Total Costs ^c^	I	11,710	10,045	16,136	8400	10,061	7163	12,610	9022	10,643
	II	14,338	18,588	9696	13,722	13,893	10,537	18,287	12,340	13,925
	III	18,501	20,283	16,080	13,540	15,601	13,506	20,465	18,084	17,007
	IV	21,997	26,381	17,677	14,074	19,573	18,367	19,852	16,909	19,354
	NA	7339	18,648	9776	8593	14,588	11,652	13,323	8426	11,543

^a^ Pool of Cancer Registries, ^b^ Prevalent cases excluding short-term survivors, ^c^ Patient annual average costs in Euros.

**Table 5 ijerph-18-00474-t005:** Prevalent cases in initial phase and treatment regimen by stage at diagnosis and age at prevalence.

Treatment Regimen	Age at Prevalence	Stage at Diagnosis					Total
		I	II	III	IV	X	
Prevalent cases ^a^	15–49	10	22	33	19	14	98
	50–69	163	180	204	65	124	736
	70–79	100	164	118	41	100	523
	80+	46	73	63	11	54	247
Patients (%) receiving surgery treatment ^b^	15–49	90	91	94	84	86	90
	50–69	88	92	96	85	73	88
	70–79	79	96	97	80	72	87
	80+	80	96	98	91	54	84
Patients (%) receiving chemo-therapy ^b^	15–49	10	55	64	68	43	56
	50–69	11	29	66	82	43	43
	70–79	7	21	47	54	33	30
	80+	2	4	13	9	6	6
Patients (%) receiving neoadjuvant chemo-therapy ^c^	15–49	10	32	18	16	57	26
	50–69	7	18	12	22	26	15
	70–79	5	4	8	12	17	8
	80+	0	0	0	0	1.9	0.4
Patients (%) receiving radio-therapy ^b^	15–49	10	41	42	32	57	40
	50–69	15	35	40	22	37	31
	70–79	12	26	33	24	38	27
	80+	22	16	13	0	19	14
Patients (%) receiving neoadjuvant radio-therapy ^c^	15–49	10	14	15	11	50	18
	50–69	10	18	11	11	21	14
	70–79	8	10	13	7	13	11
	80+	11	7	2	0	9	6

^a^ Prevalent cases excluding short-term survivors, ^b^ % values are computed over all patients in initial phase, ^c^ % values are computed over patients with surgery in initial phase.

## Data Availability

The data presented in this study are available on request from the corresponding author. The data are not publicly available yet as the trial is currently ongoing.

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
