# Peer review of "The Economic Impact of Rectal Cancer: A Population-Based Study in Italy"

_ijerph, 2021, doi:10.3390/ijerph18020474_

Round 1

Reviewer 1 Report

This article begins with a thoughtful review of important characteristics of Italy's health care system and rectal cancer care.  For those who have interests in health and cancer care cost it is an effective reminder in high-income countries and some developing countries .  Thus, the initial review is relevant to other places.

In lines 169-170, if  a patient dies for causes other than cancer in this study,  the case is assigned to the initial or continuing phase of care. It is a pity that I do not understand. To my knowlege,  a cancer can result in  other diseases or cancer. The patient dies for other cancers or disease, how do you assure yourself that the assigned cases are accurately reported.  Could you give me more information?

In lines 187-189,it might be useful to have some clarity about the patient monthly average cost, such as hosipital cost, outpatient service cost and costs due to drugs prescriptions. In additional,  the mode value, median value, average value of hosiptal admissions, outpatient services and drug prescriptions should be offered.

In line 216,  I am assuming that this is a self-report? Because I can not find the data in Table 1. And in Table 1,  some abnormal values could be given more eyes. For example,  the average value of hospital admissions in Napoli is 2.1, higher than other areas. This is an interesting issue. I wish that more details or contexts can be offered in the revised manuscript.

Some of the findings are to be expected.  I would anticipate that there are characteristics of Italy's health care system that impact findings.

In Figure 1 and lines 231-236,  you report the average costs in each phase of care. However, the drug costs would be the part of  the hospital costs or the outpitent services cost,  how do you accurately identify the three costs? What's important, some cost due to supplier induced demand could exits. 

At last, the number of paragraphs in the Section Discussion is too much, I think it need to be retrenched.  

I think this paper is worth revising and strengthening with some much needed improved.

It's my pleasure to review your paper. Thanks.

Author Response

Response Letter:

Dear Editor and dear Reviewers,

Thank you very much for giving us the opportunity to revise this paper and, hopefully, to improve it. Some of the comments are common to more than one reviewer, meaning that an improvement in the explanation is necessary. In particular: we have revised figures 1 and 2, in order to make it more understandable (the new figures are provided separately as .pdf files); we have added in the appendix an extra figure to describe the periods of data used, which vary from one cancer registry to another; we have revised the methods, and pointed out the difference between the cross-sectional study design adopted here, and the longitudinal study design generally found in the literature, as well as the advantages of the cross-sectional approach; we have retrenched the Discussion and enlarged the Conclusions.

In what follows detailed answers to the reviewers’ specific comments are written in red, and new sentences to be reported in the paper are written in blue.

Reviewer 1:

This article begins with a thoughtful review of important characteristics of Italy's health care system and rectal cancer care.  For those who have interests in health and cancer care cost it is an effective reminder in high-income countries and some developing countries. Thus, the initial review is relevant to other places.

Q1

In lines 169-170, if a patient dies for causes other than cancer in this study, the case is assigned to the initial or continuing phase of care. It is a pity that I do not understand. To my knowlege, a cancer can result in other diseases or cancer. The patient dies for other cancers or disease, how do you assure yourself that the assigned cases are accurately reported.  Could you give me more information?

R1

We added the following sentence to lines187-191: “Causes of death are classified according to International Classification of Diseases Tenth Revision (ICD-10). Causes of death other than cancer are: S00–T98 (injury, poisoning and certain other consequences of external causes such as burn, frostbite, etc.),V01–Y98 (external causes of morbidity and mortality, such as transport accident, drowning, exposure to forces of nature, etc.).”

These causes are surely not related to the cancer under study, and if included the patients who die for these causes in the final phase they would induce some bias

Q2

In lines 187-189,it might be useful to have some clarity about the patient monthly average cost, such as hosipital cost, outpatient service cost and costs due to drugs prescriptions.

R2

Patient monthly average cost and all cost indicators are calculated separately for each health care service. We modified the following sentence in line 206: “The indicators below are computed:” with “The indicators below are computed separately for each health care service:”

Q3

In additional,  the mode value, median value, average value of hosiptal admissions, outpatient services and drug prescriptions should be offered.

R3

Table 1 describes the whole cohort, which is largely made by patient in the continuing phase (73% who were diagnosed between 2 and 8 years before), therefore the mode and median events are zero. The average value is the most appropriate to provide a picture of the different cancer registries.

Q4

In line 216,  I am assuming that this is a self-report? Because I can not find the data in Table 1.

R4

In fact, the number of person-years is an irrelevant detail that might generate confusion, and it has been deleted. See line 240.

Q5

And in Table 1,  some abnormal values could be given more eyes. For example, the average value of hospital admissions in Napoli is 2.1, higher than other areas. This is an interesting issue. I wish that more details or contexts can be offered in the revised manuscript.

R5

Napoli cancer registry was enlarged during the period of observation and part of the population of the CR was observed only for a three-year period. This implies that the proportion of newly diagnosed cases is higher: prevalent cases in initial phase in Napoli are more frequent (25%) than in the other Cancer Registries (on average 18%). This fact was originally stressed in the Discussion section, and now it is anticipated in the Results section (from lines 425-429 of previous version to lines 246-249): “Notice that part of the population of Napoli CR was covered with only 3 years of registration, hence the study cohort is characterized by a higher proportion of newly diagnosed patients, who require more treatments and hospital admissions, and by a lower proportion of intermediate patients. "

Q6

Some of the findings are to be expected.  I would anticipate that there are characteristics of Italy's health care system that impact findings.

R6

The main characteristics of the Italian health care system, including management at regional level, are reported in the Data sources section: see lines 100-109.

Q7

In Figure 1 and lines 231-236, you report the average costs in each phase of care. However, the drug costs would be the part of the hospital costs or the outpatient services cost, how do you accurately identify the three costs? What's important, some cost due to supplier induced demand could exits. 

R7

The following sentence is included in lines 142-152: Drugs may be administered to patients in three settings: hospital, outpatient clinic, pharmacy. The DP database contains detailed information (including molecule and corresponding ATC) only on drugs prescribed to patients and sold by pharmacies. The OPS database includes generic information on chemotherapy drugs administered in outpatient (not molecule nor ATC). The HD database includes the cost of drugs administered during hospital stay in the DRG system, which assigns an overall reimbursement for treatments, procedures, interventions, drugs, and does not contain detailed information (not molecule nor ATC).

Finally, high cost drugs, such as biological drugs, MABs, etc. are included in a different database which was not used in our study, because during data collection we discovered that the information processing and the refund system was widely variable, in terms of completeness, from Region to Region. In conclusion, detailed information on costs of drugs is available for drugs sold by pharmacies, only.”

This sentence substitutes lines 140-143 of previous version (deleted lines 153-157): Cost information on very expensive drugs administered in hospital, such as immune-therapeutic drugs, are in another database not included in the analysis, because during data collection we discovered that the information processing and the refund system was widely variable, in terms of completeness, from Region to Region. However, at that time high cost drugs were scarcely administered in Italy.”

Q8

At last, the number of paragraphs in the Section Discussion is too much, I think it need to be retrenched. 

R8

The Discussion section has been fully revised, some sentences were deleted, other sentences were moved to the Conclusions. In details:

Sentence: “To our knowledge, this paper is the first study to estimate, at population level using micro-data, the economic burden of rectal cancer over the public health system in Italy. Estimation is based on a three-phase pattern of care, that considers the whole process of the disease, from initial diagnosis to cure/death. Information at individual level comes from various health care and administrative databases.” has been moved from lines 363-367 to the Conclusions (lines 487-491)

Sentence: “Patterns of care following a rectal cancer diagnosis differ from those following a colon diagnosis and affect results on costs: for example, in comparison with colon cancer patients, rectal cancer patients are treated more often with radiotherapy, and less often with chemotherapy.” (lines 369-372) has been deleted

Sentence: “It must be noticed that at the time of data collection (2009-2011 for most cancer registries) the use of hospice for terminal patients was not routinely implemented and most patients died in hospital.” (lines 382-384) has moved down to lines 428-429

Sentence: “The diffusion of organized screening programs for colorectal cancer in Italy varies by Region: in 2010, 65% of the target population was invited to the screening, with percentages ranging from 90% in Northern Regions to 24% in Southern Regions; adherence ranged from 49% of invited persons in the North, 45% in the Centre and 33% in the South of Italy. [34] This study shows that early diagnosis of rectal cancer is again on the healthcare budget; therefore, policies raising the spreading and adherence to screening plans, above all when addressed to people living in the South of Italy, should be strongly encouraged.” (lines 392-399) has been moved to the Conclusions and modified to: “Presently, the diffusion and adherence of organized screening programs for colorectal cancer in Italy is very variable among regions”. (lines 498-499)

Sentence: “Age is another determinant of patterns of care and costs: younger patients (ages 15-49) cost on average 50% more than elderly patients (age 80 and over) in initial phase, twice as much in continuing phase and four-fold in their end-of-life. These results are partially caused by different clinical approaches, as confirmed in Table 5: more aggressive (and more expensive) treatments are better tolerated by young patients, who have higher life expectancy when faced with aggressive treatments, in comparison with older patients, who generally have more co-morbidities. Further, some elderly patients live in elderly homes, whose costs database was not considered in the analysis.” (lines 400-407) has been modified to: “Age is another determinant of costs since clinical approaches vary by age: more aggressive (and more expensive) treatments are better tolerated by young patients, who have higher life expectancy when faced with aggressive treatments, in comparison with older patients, who generally have more co-morbidities".

Sentence: “There was no selection concerning prognosis or regarding any patient’s demographic and clinical feature.” has been moved from lines 416-417 to lines 410-411.

Sentence: “Data are linked on an individual basis to a cohort of prevalent cases selected by the CR. By the use of an approach that directly identifies only procedures related to cancer, only cancer care costs are produced.” (lines 412-414) has been deleted

Sentence: “where incidence cases are followed up in time” (line 416) has been deleted

Line 418: the word “follow up” has been added

Line 429: the sentence “However, it must be noticed that” has been added

Sentence: “This incompleteness might limit the comparability of initial phase costs between cancer registries.” Has been added (lines 433-434)

Sentence: “Unknown stage at diagnosis is often associated to patients with more severe illness: when metastases occur, the patient may not receive a surgery and therefore information on biopsy may be missing, and consequently stage may not be reported in the CR files. In this study 8.5% of patients with unknown stage do not undergo surgery: these are certainly metastatic patients, who should be added to the 5.5% stage IV patients. Furthermore, some misclassification of more severe stages may occur: for example, suppose a patient undergoes surgery and is classified in stage III; in case subsequently metastases are detected via computed tomography scan but results are not recorded in the Hospital Discharge record, information on the CR files is not correctly reported.” (lines 435-443) has been deleted.

Sentence: “This data source will hopefully improve in completeness and standardization in the next few years and specific data check procedures were developed in the Epicost study, in order to be included in another call for data.” (lines 448-450) has been modified and moved to the Conclusions: “Standardization and completeness of in-hospital drugs database has improved in more recent years. In a feature perspective, specific data check procedures developed in the Epicost study will be used to include in-hospital drugs database in cost analysis.” (lines 500-502)

Sentence: “In a further development, homogeneous groups of patients in terms of similar care needs will be identified through specific procedures [36].” (lines 454-455) has been modified and moved to the Conclusions: “Furthermore, in the continuing phase, homogeneous groups of patients in terms of similar care needs will be identified through specific procedures [37].” (lines 502-504).

Sentence: “We presented results by region because in Italy the reimbursement system is regionally based. “ has been moved from lines 471-472 to line 456.

Sentence: “There are some confounders that may impact on the comparison of results at geographic level:” (line 457-458) has been modified: “However, regional comparison of total costs by phase of care is limited by the following confounders:”

Sentence: “The usage of hospice, instead of hospitalization, is largely variable among the CR area, affecting results comparability in final phase of the disease.” (lines 459-460) has been deleted

Sentence: “We identified two situations that require specific comments: Part of the population of Napoli CR was covered with only 3 years of registration, hence the study cohort is characterized by a higher proportion of newly diagnosed patients, who require more treatments and hospital admissions, and by a lower proportion of intermediate patients. This explains the observed overall higher rates of hospitalizations, outpatient services and drug prescriptions in this CR.” (lines 476-481) has been modified and moved to Results section (lines 246-249): “Notice that part of the population of Napoli CR was covered with only 3 years of registration, hence the study cohort is characterized by a higher proportion of newly diagnosed patients, who require more treatments and hospital admissions, and by a lower proportion of intermediate patients.”

I think this paper is worth revising and strengthening with some much needed improved.

It's my pleasure to review your paper. Thanks.

Reviewer 2 Report

I read with interest the manuscript numbered ijerph-988080. I hope my English is sufficiently clear as to make my suggestions understandable.  

Abstract

I think that the Abstract must contain the calendar years or the time period to which the cost data pertain.

91-92 and elsewhere. “…linking administrative healthcare sources to cancer registry’s source)”.

It is unclear to me (unless I have overlooked something) whether the authors have used local (institutional) or region-wide administrative data sources. This is a key information. 

127-128. “Each service is coded according to the ICD9-CM coding system, however each Region sets its cost, and decides whether to add more codes corresponding to extra services not included in the ICD9 claim list”.

Having taken the regional extra codes into account is a strength of the study. They may have a significant impact on total costs. In order to help other researchers to replicate the study in other Italian regions, I suggest the authors to provide, in an appendix, the ICD9-CM codes they have used. This would be necessary for the ATC codes too.

  1. 2.2. Study cohorts

(Incidentally: why cohorts is used in the plural?). Did the cohort include immigrants? This could introduce a bias into results linked to services delivered to patients abroad.   

  1. “… an anonymous personal identification alpha-numeric code …”.

I suggest the authors to provide some more details and some clarifications on this code. Administrative data sources have a 8-digit numeric code which – I believe – has nothing to do with the codes used by the registries. The only alpha-numeric code I know is the fiscal code, which is not an anonymous identifier.        

177-178. “Only events related to rectal cancer are considered in a list of correlated events, one for each database. Lists are elaborated by oncologists and clinicians …”.

Based on which criteria was an event defined as being rectal-cancer-related? 

  1. 4. Results

The authors have evaluated the events occurring in 12 months of the clinical history (in different phases of the disease) of patients. The total costs of these events have then been plotted on a schematic or virtual representation of the clinical history of rectal cancer, that is depicted in the horizontal axis of Figure 1. If I understand correctly the study design, however, they have in their dataset a much longer period of observation. Can the authors provide some summary information on the actual length of the clinical history of patients?   

Figure 1 (page 7). I like it, but there are some problems. (1) The name of the two vertical axes are difficult to read. Perhaps the Figure may be tightened and the names written in a larger font. (2) More importantly, the legends under the horizontal axis are placed in a misleasing manner. In the greater part of the initial phase, as it stands now, the cost of hospitalization is less than the cost of outpatient services. This is also the case for the final phase. This contrasts with the statement (line 279) that “Hospitalization costs account for 85% and 83% of total costs in initial and final phases”, wich I believe is correct. There is a graphical problem with the legends under the horizontal axis. The initial and final phases are probably much shorter than depicted in the Figure. (3) Closely related, it is unclear to me what the tick marks in the horizontal axis indicate. Months? This is impossible. Virtual time units? If so, they can be misleading. Perhaps the authors may consider the opportunity of delete the tick marks, because the time – as represented in the horizontal axis – is “immaterial”. (4) Another imperfection is that, even in the very early and very final phases, hospitalization costs do not seem to account for over 80% of total costs.                 

Figure 2 (page 8).

The above error in Figure 1 (the length of the initial and final phases) makes Figure 2 be meaningless. In the upper panel, the sum of the three items of expenditure in the continuing phase accounts for most of total costs. I think that this is true – but is not confirmed by current Figure 1. Only if the continuing phase in Figure 1 enlarges substantially in both directions, it would appear that most of the cost for rectal cancer care is for the continuing phase. Incidentally, some of the colours in the small boxes in the panel (a) are difficult to distinguish from each other. I guess the boxes are too small.                    

Line 395. “In-hospital drugs database is not considered in the analysis, because the archives were incomplete and of poor quality”.

This statement is unclear to me. In-hospital drug databases were created in 2007 and the data have been usable since 2008. Have the authors encountered some specific problem? If so, it would be useful to point them out. 

339-340. “… the use of hospice for terminal patients was not routinely implemented and patients died in hospital”.

This may be true. In the study years, this was still the case for the greater part of Italy (I checked it out). Consequently, the lack of data for the cost of care of terminally-ill patients cannot be a major cause of the (likely) underestimate of total costs. See next comment, too.           

378-379. “Some data sources are not considered in this study: home care services, nursing facilities for elderly people, emergency room (ER) services, hospices for terminal patients. As a consequence, total costs might be underestimated”.

Based on other similar studies (if the authors consider them comparable to their own), what could be the magnitude of the underestimate? That is, can the authors provide a rough estimate of the percentage of total cost that is accounted for by the above items of expenditure? A related question could be: do the authors agree that in-hospital drug provision are a major – if not the most important – item of expenditure in rectal cancer care for which the data were only partially available?   

Author Response

Response Letter:

Dear Editor and dear Reviewers,

Thank you very much for giving us the opportunity to revise this paper and, hopefully, to improve it. Some of the comments are common to more than one reviewer, meaning that an improvement in the explanation is necessary. In particular: we have revised figures 1 and 2, in order to make it more understandable (the new figures are provided separately as .pdf files); we have added in the appendix an extra figure to describe the periods of data used, which vary from one cancer registry to another; we have revised the methods, and pointed out the difference between the cross-sectional study design adopted here, and the longitudinal study design generally found in the literature, as well as the advantages of the cross-sectional approach; we have retrenched the Discussion and enlarged the Conclusions.

In what follows detailed answers to the reviewers’ specific comments are written in red, and new sentences to be reported in the paper are written in blue.

Reviewer 2:

I read with interest the manuscript numbered ijerph-988080. I hope my English is sufficiently clear as to make my suggestions understandable.   

Q1

Abstract

I think that the Abstract must contain the calendar years or the time period to which the cost data pertain.

R1 

This is a good point. We added the following sentence in lines 41-42: “Prevalence date varies among CRs and spans from 2009 (January 1st) to 2013 (January 1st). Cost estimates for the pool of CRs refer mainly to years 2010-2011.”

Q2

91-92 and elsewhere. “…linking administrative healthcare sources to cancer registry’s source)”.

It is unclear to me (unless I have overlooked something) whether the authors have used local (institutional) or region-wide administrative data sources. This is a key information. 

R2

we pointed out that the administrative data sources are regional-based, by modifying sentence in line 91 of previous version: “(linking administrative healthcare sources to cancer registry’ s source)” to:“(linking administrative regional-based healthcare sources to cancer registry’ s source)”.(see line 93)

Q3

127-128. “Each service is coded according to the ICD9-CM coding system, however each Region sets its cost, and decides whether to add more codes corresponding to extra services not included in the ICD9 claim list”.

Having taken the regional extra codes into account is a strength of the study. They may have a significant impact on total costs. In order to help other researchers to replicate the study in other Italian regions, I suggest the authors to provide, in an appendix, the ICD9-CM codes they have used. This would be necessary for the ATC codes too.

R3 

Lists of colorectal cancer-related codes (ICD9-CM and ATC)are described in a paper which is about to be submitted for publication in the same special issue the present manuscript is submitted, therefore could not be shown here.

Q4

  1. 2. Study cohorts

(Incidentally: why cohorts is used in the plural?). Did the cohort include immigrants? This could introduce a bias into results linked to services delivered to patients abroad.   

R4

The wording “cohorts” has been changed into “cohort” throughout the text. See lines 158, 165, 551.

The cohort includes all residents in the areas covered by cancer registration, who have been treated in Italy, regardless of their nationality, as the Italian system is universal, i.e. it provides services to all residents. Services provided abroad and not reimbursed by the Italian National Health Care System are not included in the analysis. This argument has been treated in lines 100-109.

Q5

  1. “… an anonymous personal identification alpha-numeric code …”.

I suggest the authors to provide some more details and some clarifications on this code. Administrative data sources have a 8-digit numeric code which – I believe – has nothing to do with the codes used by the registries. The only alpha-numeric code I know is the fiscal code, which is not an anonymous identifier.     

R5

To clarify: Cancer registries are authorized to see the nominative data of cancer patients (according to the Regional Regulations approved by the Italian Data Protection Authority). The registries firstly link cancer data to administrative sources using the fiscal code, then anonymize all information using an anonymous unique identification code for each patient and provide the data to the analysis centre. We do not think it necessary to add this clarification into the text.

 Q6

177-178. “Only events related to rectal cancer are considered in a list of correlated events, one for each database. Lists are elaborated by oncologists and clinicians …”.

Based on which criteria was an event defined as being rectal-cancer-related? 

R6

The lists were drawn by a multidisciplinary panel of oncologists and cancer registries experts and are based on a combination of guidelines and practical protocols implemented in Italy. We modified the sentence in lines 199-202: “Lists are elaborated by oncologists and clinicians and comprise…” to: “Lists are elaborated by oncologists and clinicians on the basis of clinical guidelines and current practice and comprise…”

 Q7

  1. Results

The authors have evaluated the events occurring in 12 months of the clinical history (in different phases of the disease) of patients. The total costs of these events have then been plotted on a schematic or virtual representation of the clinical history of rectal cancer, that is depicted in the horizontal axis of Figure 1. If I understand correctly the study design, however, they have in their dataset a much longer period of observation. Can the authors provide some summary information on the actual length of the clinical history of patients?   

R7

The CR provides the date of diagnosis and clinical information such as cancer topography and morphology and life status follow up. Information on costs is provided by the CR for a period of 24 months, centred around prevalence date. For each patient costs are estimated in a 12-month period, centred around prevalence date, as explained in appendix. In general, there are two possible study designs for a study like this:

1) a longitudinal (or incidence-based) approach: an incident cohort of patients (i.e. diagnosed in a given period) is followed over a certain number of years; during this interval of time a single patient may go through different phases of care. This is the approach more widely used to estimate costs of a disease (see for example Mariotto, 2011; Francisci, 2013; Laudicella, 2016)

2) a cross-sectional (or prevalence-based) approach: different incident cohorts (i.e. diagnosed in different periods) are considered and are studied in a given interval of time (usually 12 months) centred around a fixed prevalence date; in the period of study a single patient belongs to one phase of care, only. This approach has been successfully applied by demographers to estimate population life expectancy and in cancer descriptive epidemiology to provide up-to-date estimate of cancer survival from population-based cancer registry data (see for example Brenner and Gefeller, J Clinical Epidem 1997; Brenner, Lancet 2002; Brenner and Hakulinen, American J Epidemiol 2002).

In this study we used a cross-sectional study design (similarly to Xue Qin Yu in PlosOne 2017), i.e. we looked at costs in a 12-months period centred around the prevalence date. This period has a different starting and ending point for each patient, as illustrated in the appendix. The reasons for this choice are the following: a) it provides more updated estimates of costs, in particular for the initial phase of care. In the case of the longitudinal approach cost estimates of the initial phase refer to patients diagnosed many years before and this limits the purpose of the analysis; b) cross-sectional cost data are directly collected at constant price; as a consequence, there is no need for temporal price index to transform nominal into real values; c) embodied and/or disembodied technological progress affects the type of treatments as well as the way they are administered and, consequently, the cost profile. Cross-sectional data are better suited than longitudinal data to capture the effects of technological change.

We modified the Results, Methods and Appendix sections in the following way:

Results:

Sentence: “This study involves 8 population-based CRs having a minimum of 8 years of cancer registration: Milano, Friuli Venezia Giulia (VG), Veneto in the North; Firenze-Prato, Umbria, Latina in the Centre; Palermo, Napoli in the South; overall, they cover just over 10 million people, corresponding to about one sixth of the Italian population. Each CR uses the most updated data at the time of case extraction: the study cohort includes patients diagnosed with malignant rectal cancer (ICD9-CM C19, C20) in the most recent 8 years of incidence and still alive at prevalence date (prevalence cohorts). Persons who were previously diagnosed of a cancer in the five years before diagnosis of rectal cancer, or persons diagnosed with a further cancer in the year after diagnosis of rectal cancer, were excluded. Prevalent cases are followed up to one year after prevalence date, with respect to their vital status. Prevalence date varies among CRs and is between 2009 (January 1st) and 2013 (January 1st); incidence periods contributing to the analysis span between 2001-2008 to 2005-2012 (Table 1).” has been modified to: “This study involves 8 population-based CRs having a minimum of 8 years of cancer registration: Milano, Friuli Venezia Giulia (VG), Veneto in the North; Firenze-Prato, Umbria, Latina in the Centre; Palermo, Napoli in the South; overall, they cover just over 10 million people, corresponding to about one sixth of the Italian population. We use a cross-sectional study design: the study cohort includes patients diagnosed with malignant rectal cancer (ICD9-CM C19, C20) in the most recent 8 years of incidence and still alive at prevalence date (prevalence cohort), as illustrated in Table 1. Each CR uses the most updated data at the time of case extraction, thus prevalence date varies among CRs: from 2009 (January 1st) to 2013 (January 1st). In each CR administrative data used for cost analysis is available for a 24-month period centred around prevalence date. Persons who were previously diagnosed of a cancer in the five years before diagnosis of rectal cancer, or persons diagnosed with a further cancer in the year after diagnosis of rectal cancer, were excluded. Prevalent cases are followed up to one year after prevalence date, with respect to their vital status.” (lines 159-172)

Methods:

The following sentence has been added in lines 195-196: “Figure A2 in appendix illustrates the periods when data for cost analysis is available for each CR.”

Appendix:

we added a new paragraph to illustrate in details the periods of data available for cost analysis in each cancer registry (lines 573-585): “The study cohort includes patients with malignant rectal cancer diagnosis (ICD9-CM C19, C20) during the last 8 years of incidence in the areas covered by the 8 Cancer Registries contributing to the study and still alive at prevalence date (prevalence cohort). Each registry uses the most up-to-date data available at the time of case retrieval. In details:

  • period of incidence in Firenze CR is 2001-2008; prevalence date is January 1st2009; data for cost analysis is available in period 1.1.2008 to 31.12.2009;
  • period of incidence in Veneto and Friuli VG CRs is 2002-2009; prevalence date is January 1st2010; data for cost analysis is available in period 1.1.2009 to 31.12.2010;
  • period of incidence in Latina, Napoli, Palermo and Umbria CRs is 2003-2010; prevalence date is January 1st2011; data for cost analysis is available in period 1.1.2010 to 31.12.2011;
  • period of incidence in Milano CR is 2005-2012; prevalence date is January 1st2013; data for cost analysis is available in period 1.1.2012 to 31.12.2013.

Figure A2 illustrates the periods when data for cost analysis is available in each CR.”

Q8

Figure 1 (page 7). I like it, but there are some problems. (1) The name of the two vertical axes are difficult to read. Perhaps the Figure may be tightened and the names written in a larger font. (2) More importantly, the legends under the horizontal axis are placed in a misleasing manner. In the greater part of the initial phase, as it stands now, the cost of hospitalization is less than the cost of outpatient services. This is also the case for the final phase. This contrasts with the statement (line 279) that “Hospitalization costs account for 85% and 83% of total costs in initial and final phases”, wich I believe is correct. There is a graphical problem with the legends under the horizontal axis. The initial and final phases are probably much shorter than depicted in the Figure. (3) Closely related, it is unclear to me what the tick marks in the horizontal axis indicate. Months? This is impossible. Virtual time units? If so, they can be misleading. Perhaps the authors may consider the opportunity of delete the tick marks, because the time – as represented in the horizontal axis – is “immaterial”. (4) Another imperfection is that, even in the very early and very final phases, hospitalization costs do not seem to account for over 80% of total costs.

R8

Figure 1 has been redrawn, as having two scales (hospitalization to the left-hand side and outpatient services to the right-hand side) was confusing:

  1. we split cost profiles due to hospitalization and due to outpatient services into two separate plots; drugs costs are so small in comparison to the other two sources, that are not interesting to show; notice that scales are different and hospitalization scale is much higher
  2. we added labels in X-axis: the first 12 labels refer to the 12 months of the initial phase, the second 12 labels to the 12 months of the continuing phase, the third 12 labels to 12 months of the final phase, as illustrated in the Methods section (see line 220). We clarified in the answer to Q7 that each phase (including the continuing phase) comprises costs over 12 months.

We modified the description of Figure 1 as follows:

Old caption: “Figure 1. Monthly average cost profiles (€) by type of service: hospitalization (LHS scale) and outpatient (RHS scale). Pool of Cancer Registries.”

New captions:

”Figure 1a. Cost profile (or patient monthly average costs) due to hospitalization. Pool of CRs"

Figure 1b. Cost profile (or patient monthly average costs) due to outpatient services. Pool of CRs.”(lines 263-264)

Old text: “The X-axis measures the time in each phase of care: from diagnosis to 12 months after it in the initial phase; from 6 months before prevalence date to 6 months after prevalence date in the continuing phase; from the 12th to the last month before death in the final phase. The Y-axis measures the average monthly cost per patient, the left-hand side scale is for hospitalization, the right-hand side scale is for outpatient. The continuous line represents the cost of hospitalization, the dotted line the cost of outpatient services.”

New text: “Cost estimates for the pool of CRs refer mainly to years 2010-2011.The X-axis measures the time in each phase of care: I1,…, I12indicate the 12 months of initial phase; C1,…, C12the 12 months of continuing phase; F1,…, F12the 12 months of final phase. The Y-axis measures the monthly average cost per patient.” (lines 257-260)

Q9

Figure 2 (page 8).

The above error in Figure 1 (the length of the initial and final phases) makes Figure 2 be meaningless. In the upper panel, the sum of the three items of expenditure in the continuing phase accounts for most of total costs. I think that this is true – but is not confirmed by current Figure 1. Only if the continuing phase in Figure 1 enlarges substantially in both directions, it would appear that most of the cost for rectal cancer care is for the continuing phase. Incidentally, some of the colours in the small boxes in the panel (a) are difficult to distinguish from each other. I guess the boxes are too small.                    

R9

Figure 1 describes the patient monthly averagecosts, as defined in the Methods (see line 209). Figure 2 describes the distribution of the total annual costs, as defined in the Methods (see line 221), by phase of care and type of service. Figure 2 has been redrawn, so as to group each phase with a different colour (initial phase = green, continuing phase = blue, final phase = red), and each type of service with a different shade of colour (drugs costs = darker, outpatient costs = lighter, hospitalization costs = medium).

We modified the caption and description of Figure 2 as follows:

Old caption: “Figure 2. Costs (a) and prevalent cases (b) distribution by type of service and phase of care.”

New caption (lines 285-286): “Figure 2. Distribution of total annual costs (a) and prevalent cases (b) by type of service and phase of care. Pool of cancer registries.”

Old sentence: “Figure 2 shows the distribution of costs by type of service in each phase of care (a) and of prevalent cases (b), for the pool of cancer registries”

New sentence (lines 280-282): “Figure 2 shows the distribution of patient annual average costs by type of service in each phase of care (a) and the distribution of prevalent cases (b), for the pool of cancer registries. Cost estimates refer mainly to years 2010-2011.”

Q10

Line 395. “In-hospital drugs database is not considered in the analysis, because the archives were incomplete and of poor quality”.

This statement is unclear to me. In-hospital drug databases were created in 2007 and the data have been usable since 2008. Have the authors encountered some specific problem? If so, it would be useful to point them out. 

R10

We did collect data from the in-hospital drug database (the so-called “File F”), however we found the database largely incomplete and not reliable. To give you an example, the same drug administered to the same patient in two different occasions was recorded once with the price of the single dose and another time with the price of the entire pack, which may contain more doses. The two values were not comparable and we did not have any way to normalize prices, therefore we decided not to use this source of information. We reported this information in the Materials section, which has been modified as follows:

Old sentence: “Cost information on very expensive drugs administered in hospital, such as immune-therapeutic drugs, are in another database not included in the analysis, because during data collection we discovered that the information processing and the refund system was widely variable, in terms of completeness, from Region to Region. However, at that time high cost drugs were scarcely administered in Italy.”

New sentence: “Drugs may be administered to patients in three settings: hospital, outpatient clinic, pharmacy. The DP database contains detailed information (including molecule and corresponding ATC) only on drugs prescribed to patients and sold by pharmacies. The OPS database includes generic information on chemotherapy drugs administered in outpatient (not molecule nor ATC). The HD database includes the cost of drugs administered during hospital stay in the DRG system, which assigns an overall reimbursement for treatments, procedures, interventions, drugs, and does not contain detailed information (not molecule nor ATC). Finally, high cost drugs, such as biological drugs, MABs, etc. are included in a different database which was not used in our study, because during data collection we discovered that the information processing and the refund system was widely variable, in terms of completeness, from region to region. In conclusion, detailed information on costs of drugs is available for drugs sold by pharmacies, only."(lines 142-152)

This is a pity, as high cost drugs are nowadays the largest share of the healthcare costs. In a future perspective this database will be taken into account, as pointed out in the Conclusions (see lines 500-502)

Q11

339-340. “… the use of hospice for terminal patients was not routinely implemented and patients died in hospital”.

This may be true. In the study years, this was still the case for the greater part of Italy (I checked it out). Consequently, the lack of data for the cost of care of terminally-ill patients cannot be a major cause of the (likely) underestimate of total costs. See next comment, too.           

R11

We agree with this comment, and indeed we do not consider the lack of hospice data a major cause of underestimation of costs.

Q12

378-379. “Some data sources are not considered in this study: home care services, nursing facilities for elderly people, emergency room (ER) services, hospices for terminal patients. As a consequence, total costs might be underestimated”.

Based on other similar studies (if the authors consider them comparable to their own), what could be the magnitude of the underestimate? That is, can the authors provide a rough estimate of the percentage of total cost that is accounted for by the above items of expenditure?

R 12

This type of study, based on microeconomic data, is innovative in Italy, and to our knowledge there are not similar studies. There are several international studies (that have been cited in the bibliography) but cost comparison is very hard, as healthcare systems as well as reimbursement systems are very variable among nations. Altini et al in a recently published paper (IJERPH 2020) on costs of gastrointestinal cancers in the area of Forlì-Cesena (North Italy) in 2016 allocate 6.6% of total costs to hospice and home care, and 0.9% to ER services.

Q 13

A related question could be: do the authors agree that in-hospital drug provision are a major – if not the most important – item of expenditure in rectal cancer care for which the data were only partially available? 

R13  

On the basis of the information contained in the in-hospitaldrug database (File F) we see that at the time of data collection in-hospital drugs accounted to 15.1% of total costs. However, we do not feel confident to use these data, for the reasons explained in R10. Notice that Altini et al allocate 12% of total costs to in-hospital drugs.

Reviewer 3 Report

Referee report for the manuscript entitled “The economic impact of rectal cancer: a population-2 based study in Italy”

This is an interesting manuscript dealing with a very important topic, the economic impact of one of the cancer with highest prevalence. It is in good shape and well written although the edition should be revised as there are some minor typos or incomplete sentences.

My main comment is that the design of the study should be better explained in the main text, although the appendix is quite useful. Specifically, it was difficult to me to understand how the prevalence date is defined at each site so that the three different stages at which a patient can be assigned are conformed.

Also, if different sites are using a different prevalence date, the evolution of prices should be taken into account. Maybe that is the reason why some sites are more expensive than others? (prices usually grow in time). The other reason is that some sites might have in general greater prices than others. If that is not the case, then, there might be a problem with the sample with greater severity in patients of the expensive sites, or different protocols or clinical guides.

In figure 1, it is not clear which serie represents hospitalization costs, and which one represents outpatient and pharmaceutical expenditures (are both together? why not separately?). Only in the text in page 7 it is stated what the dotted line and the continuous line are. This information should be present in the figure.

Given that the information seems to be available in the dataset the authors are using, I would suggest to use a smaller sample but with the entire information for each patient, with the entire profession of the disease. That means, taking patients in their initial states and following them until the last stage (for them, which might be the continuing or the final stage). That way, any individual heterogeneity would be taken into account. Also, that would provide important information about the duration or length of the second stage (continuing).

The discussion section would be benefitted from a deeper comparison of the results of this paper with others from the literature.

With respect to limitations, the authors state that different sites are using resources in different ways (for instance, hospitalizations or outpatient). Are those differences enough to invalidate the analysis of the cost per resource? should we just look to the total cost? A different limitation is that all pharmaceutical expenditure should be included, either as inpatient pharmaceutical expenditure or as outpatient expenditure. Specifically in the case of cancer patients, pharmaceutical expenditures may be large, and if there is a proportion of them excluded from the study, it should be stated how large that proportion is.

Also, the conclusion section is very short. I would suggest the authors to add further comments on policy implications derived from their results.

Author Response

Response Letter:

Dear Editor and dear Reviewers,

Thank you very much for giving us the opportunity to revise this paper and, hopefully, to improve it. Some of the comments are common to more than one reviewer, meaning that an improvement in the explanation is necessary. In particular: we have revised figures 1 and 2, in order to make it more understandable (the new figures are provided separately as .pdf files); we have added in the appendix an extra figure to describe the periods of data used, which vary from one cancer registry to another; we have revised the methods, and pointed out the difference between the cross-sectional study design adopted here, and the longitudinal study design generally found in the literature, as well as the advantages of the cross-sectional approach; we have retrenched the Discussion and enlarged the Conclusions.

In what follows detailed answers to the reviewers’ specific comments are written in red, and new sentences to be reported in the paper are written in blue.

Reviewer 3:

This is an interesting manuscript dealing with a very important topic, the economic impact of one of the cancer with highest prevalence. It is in good shape and well written although the edition should be revised as there are some minor typos or incomplete sentences.

Q1

My main comment is that the design of the study should be better explained in the main text, although the appendix is quite useful. Specifically, it was difficult to me to understand how the prevalence date is defined at each site so that the three different stages at which a patient can be assigned are conformed.

R1

We modified the Results, Methods and Appendix sections in the following way:

Results:

Sentence: “This study involves 8 population-based CRs having a minimum of 8 years of cancer registration: Milano, Friuli Venezia Giulia (VG), Veneto in the North; Firenze-Prato, Umbria, Latina in the Centre; Palermo, Napoli in the South; overall, they cover just over 10 million people, corresponding to about one sixth of the Italian population. Each CR uses the most updated data at the time of case extraction: the study cohort includes patients diagnosed with malignant rectal cancer (ICD9-CM C19, C20) in the most recent 8 years of incidence and still alive at prevalence date (prevalence cohorts). Persons who were previously diagnosed of a cancer in the five years before diagnosis of rectal cancer, or persons diagnosed with a further cancer in the year after diagnosis of rectal cancer, were excluded. Prevalent cases are followed up to one year after prevalence date, with respect to their vital status. Prevalence date varies among CRs and is between 2009 (January 1st) and 2013 (January 1st); incidence periods contributing to the analysis span between 2001-2008 to 2005-2012 (Table 1).” has been modified to: “This study involves 8 population-based CRs having a minimum of 8 years of cancer registration: Milano, Friuli Venezia Giulia (VG), Veneto in the North; Firenze-Prato, Umbria, Latina in the Centre; Palermo, Napoli in the South; overall, they cover just over 10 million people, corresponding to about one sixth of the Italian population. We use a cross-sectional study design: the study cohort includes patients diagnosed with malignant rectal cancer (ICD9-CM C19, C20) in the most recent 8 years of incidence and still alive at prevalence date (prevalence cohort), as illustrated in Table 1. Each CR uses the most updated data at the time of case extraction, thus prevalence date varies among CRs: from 2009 (January 1st) to 2013 (January 1st). In each CR administrative data used for cost analysis is available for a 24-month period centred around prevalence date. Persons who were previously diagnosed of a cancer in the five years before diagnosis of rectal cancer, or persons diagnosed with a further cancer in the year after diagnosis of rectal cancer, were excluded. Prevalent cases are followed up to one year after prevalence date, with respect to their vital status.” (lines 159-172)

Methods:

The following sentence has been added in lines 195-196: “Figure A2 in appendix illustrates the periods when data for cost analysis is available for each CR.”

Appendix:

we added a new paragraph to illustrate in details the periods of data available for cost analysis in each cancer registry (lines 573-585): “The study cohort includes patients with malignant rectal cancer diagnosis (ICD9-CM C19, C20) during the last 8 years of incidence in the areas covered by the 8 Cancer Registries contributing to the study and still alive at prevalence date (prevalence cohort). Each registry uses the most up-to-date data available at the time of case retrieval. In details:

  • period of incidence in Firenze CR is 2001-2008; prevalence date is January 1st2009; data for cost analysis is available in period 1.1.2008 to 31.12.2009;
  • period of incidence in Veneto and Friuli VG CRs is 2002-2009; prevalence date is January 1st2010; data for cost analysis is available in period 1.1.2009 to 31.12.2010;
  • period of incidence in Latina, Napoli, Palermo and Umbria CRs is 2003-2010; prevalence date is January 1st2011; data for cost analysis is available in period 1.1.2010 to 31.12.2011;
  • period of incidence in Milano CR is 2005-2012; prevalence date is January 1st2013; data for cost analysis is available in period 1.1.2012 to 31.12.2013.

Figure A2 illustrates the periods when data for cost analysis is available in each CR.”

Q2

Also, if different sites are using a different prevalence date, the evolution of prices should be taken into account. Maybe that is the reason why some sites are more expensive than others? (prices usually grow in time). The other reason is that some sites might have in general greater prices than others. If that is not the case, then, there might be a problem with the sample with greater severity in patients of the expensive sites, or different protocols or clinical guides.

R2

Thank you for this comment. The reason for choosing different prevalence dates is to use the most updated information available in each cancer registry at the time of data collection. Costs for most sites are computed in years 2010 and 2011, with exception of Firenze (representing 16% of cases whose costs are computed in 2009) and Milano (representing 28% of cases whose costs are computed in 2013). Notice that in principle costs components should be evaluated and compared at constant prices, and this implies that monetary aggregates referring to different years or regions should be adjusted to take into account price changes over time and space. However, we decided to omit this kind of adjustment basically for two reasons: i) during the time span considered (2009-2013) the temporal variation in prices in Italy was relatively low and ii) it is really difficult to identify a price index well suited for our purposes, that is a price index able to capture the prices dynamic for the specific health expenditure components we are considering in our analysis. We admit that by comparing nominal values we allow for a “monetary” bias, nevertheless, because of point i), we believe that this bias is quite limited and lower than the potential bias induced by the use of an inadequate price deflator (for instance a GDP deflator or an official consumer price index). Furthermore, as we mention in lines 401-411, ours is a real-world study, i.e. findings are at population level and there is no selection concerning prognosis or regarding any patient’s demographic and clinical feature. Finally, comparison of prices among different regions is not the main purpose of our study.

Q3

In figure 1, it is not clear which serie represents hospitalization costs, and which one represents outpatient and pharmaceutical expenditures (are both together? why not separately?). Only in the text in page 7 it is stated what the dotted line and the continuous line are. This information should be present in the figure.

R3

Figure 1 has been redrawn, as having two scales (hospitalization to the left-hand side and outpatient services to the right-hand side) was confusing:

  1. we split cost profiles due to hospitalization and due to outpatient services into two separate plots; drugs costs are so small in comparison to the other two sources, that are not interesting to show; notice that scales are different and hospitalization scale is much higher
  2. we added labels in X-axis: the first 12 labels refer to the 12 months of the initial phase, the second 12 labels to the 12 months of the continuing phase, the third 12 labels to 12 months of the final phase, as illustrated in the Methods section (see line 220). We clarified in the answer to Q7 that each phase (including the continuing phase) comprises costs over 12 months.

We modified the description of Figure 1 as follows:

Old caption: “Figure 1. Monthly average cost profiles (€) by type of service: hospitalization (LHS scale) and outpatient (RHS scale). Pool of Cancer Registries.”

New captions:”Figure 1a. Cost profile (or patient monthly average costs) due to hospitalization. Pool of CRs.

Figure 1b. Cost profile (or patient monthly average costs) due to outpatient services. Pool of CRs.”(lines 263-264)

Old text: “The X-axis measures the time in each phase of care: from diagnosis to 12 months after it in the initial phase; from 6 months before prevalence date to 6 months after prevalence date in the continuing phase; from the 12th to the last month before death in the final phase. The Y-axis measures the average monthly cost per patient, the left-hand side scale is for hospitalization, the right-hand side scale is for outpatient. The continuous line represents the cost of hospitalization, the dotted line the cost of outpatient services.”

New text: “Cost estimates for the pool of CRs refer mainly to years 2010-2011.The X-axis measures the time in each phase of care: I1,…, I12indicate the 12 months of initial phase; C1,…, C12the 12 months of continuing phase; F1,…, F12the 12 months of final phase. The Y-axis measures the monthly average cost per patient.”(lines 257-260)

Q4

Given that the information seems to be available in the dataset the authors are using, I would suggest to use a smaller sample but with the entire information for each patient, with the entire profession of the disease. That means, taking patients in their initial states and following them until the last stage (for them, which might be the continuing or the final stage). That way, any individual heterogeneity would be taken into account. Also, that would provide important information about the duration or length of the second stage (continuing).

R4

The CR provides the date of diagnosis and clinical information such as cancer topography and morphology and life status follow up. Information on costs is provided by the CR for a period of 24 months, centred around prevalence date. For each patient, costs are estimated in a 12-month period, centred around prevalence date, as explained in appendix.

In general, there are two possible study designs for a study like this:

1) a longitudinal (or incidence-based) approach: an incident cohort of patients (i.e. diagnosed in a given period) is followed over a certain number of years; during this interval of time a single patient may go through different phases of care. This is the approach more widely used to estimate costs of a disease (see for example Mariotto, 2011; Francisci, 2013; Laudicella, 2016)

2) a cross-sectional (or prevalence-based) approach: different incident cohorts (i.e. diagnosed in different periods) are considered and are studied in a given interval of time (usually 12 months) centred around a fixed prevalence date; in the period of study a single patient belongs to one phase of care, only. This approach has been successfully applied by demographers to estimate population life expectancy and in cancer descriptive epidemiology to provide up-to-date estimate of cancer survival from population-based cancer registry data (see for example Brenner and Gefeller, J Clinical Epidem 1997; Brenner, Lancet 2002; Brenner and Hakulinen, American J Epidemiol 2002).

In this study we used a cross-sectional study design (similarly to Xue Qin Yu in PlosOne 2017), i.e. we looked at costs in a 12-months period centred around the prevalence date. This period has a different starting and ending point for each patient, as illustrated in the appendix. The reasons for this choice are the following: a) it provides more updated estimates of costs, in particular for the initial phase of care. In the case of the longitudinal approach cost estimates of the initial phase refer to patients diagnosed many years before and this limits the purpose of the analysis; b) cross-sectional cost data are directly collected at constant price; as a consequence, there is no need for temporal price index to transform nominal into real values; c) embodied and/or disembodied technological progress affects the type of treatments as well as the way they are administered and, consequently, the cost profile. Cross-sectional data are better suited than longitudinal data to capture the effects of technological change. We modified the Results, Methods and Appendix sections as shown in R1

Concerning individual heterogeneity, as mentioned in lines 415-421, “We adopted a cross-sectional approach, because it produces more update results than those obtained with a longitudinal approach. Eight years of follow-up are a time interval long enough to observe the entire pattern of care, provided that in Italy a recent estimate of time to cure for colorectal cancer patients is eight years. Finally, with the phase-of-care framework all clinically significant phases of the disease are considered.”

Q5

The discussion section would be benefitted from a deeper comparison of the results of this paper with others from the literature.

R5

In the introduction, we mentioned many international studies, which can be comparable to ours in terms of methodology. However, cost comparison at international level is very hard to perform, as healthcare systems as well as reimbursement systems are very variable among nations. We could in principle compare our findings to others in Italy, but this type of study, based on microeconomic data, is innovative in Italy, and to our knowledge there are not similar studies. We found a recent paper by Altini et al, which show comparable results in terms of cost distribution. This reference has been added in line 375.

Q6

With respect to limitations, the authors state that different sites are using resources in different ways (for instance, hospitalizations or outpatient). Are those differences enough to invalidate the analysis of the cost per resource? should we just look to the total cost?

R6

The main aim of the study is to provide a snapshot of the economic impact of cancer care on the National Health Service and to this purpose we pooled together costs from different regions. Another aim is to investigate how different regional healthcare systems treat the same illness in different settings, in order to evaluate the impact of healthcare organization on the costs, as shown in Discussion (lines 482-485).

Q7

A different limitation is that all pharmaceutical expenditure should be included, either as inpatient pharmaceutical expenditure or as outpatient expenditure. Specifically in the case of cancer patients, pharmaceutical expenditures may be large, and if there is a proportion of them excluded from the study, it should be stated how large that proportion is.

R7

We did collect data from the in-hospital drug database (the so-called “File F”), however we found the database largely incomplete and not reliable. To give you an example, the same drug administered to the same patient in two different occasions was recorded once with the price of the single dose and another time with the price of the entire pack, which may contain more doses. The two values were not comparable and we did not have any way to normalize the prices, therefore we decided not to use this source of information. On the basis of the information contained in the in-hospitaldrug database we see that at the time of data collection in-hospital drugs accounted to 15.1% of total costs. However, we do not feel confident to use these data. Notice that Altini et al allocate 12% of total costs to in-hospital drugs.

Q8

Also, the conclusion section is very short. I would suggest the authors to add further comments on policy implications derived from their results.

R8

The Conclusions section has been enlarged, and 2 references, that appeared more recently, have been added.

Old version: “The approach of this study allows policy makers to identify areas with different needs: among health care services, among phases of care and among some patients’ characteristics, such as age and stage. Our model may support policy makers in predicting near future cancer burden on the basis of different scenarios induced by specific interventions. The type of analysis proposed here can be extended to other Countries with diverse healthcare managements and systems, as long as data on healthcare services and relating costs at individual level is accessible”

New version: “To our knowledge, this paper is the first study to estimate, at population level using micro-data, the economic burden of rectal cancer over the public health system in Italy. Estimation is based on a three-phase pattern of care, that considers the whole process of the disease, from initial diagnosis to cure/death. Information at individual level comes from various health care and administrative databases.

The approach of this study allows policy makers to identify areas with different needs: among health care services, among phases of care and among some patients’ characteristics, such as age and stage. Our model may support policy makers in predicting near future cancer burden on the basis of different scenarios induced by specific interventions. For example, this study shows that early diagnosis of rectal cancer is a gain on the healthcare budget; therefore, policies raising the spreading and adherence to screening plans, above all when addressed to people living in the South of Italy, should be strongly encouraged. At present, the diffusion and the adherence of organized screening programs for colorectal cancer in Italy is very variable among regions. [37]

Standardization and completeness of in-hospital drugs database has improved in more recent years. In a feature perspective, specific data check procedures developed in the Epicost study will be used to include in-hospital drugs database in cost analysis.

The type of analysis proposed here can be extended to other Countries with diverse healthcare managements and systems, as long as data on healthcare services and relating costs at individual level is accessible. As an example, in the ongoing Innovative Partnership for Action Against Cancer (iPAAC) financed by the European Commission the methodology has been proposed for application to other European countries, such as Belgium, Spain, Norway and Poland [38].“

Round 2

Reviewer 1 Report

Dear all of authors, 

    Thank you  for reviewing your paper. In this revised manuscript, you contribute to improve the article. However, the number of paragraphs in the Section Discussion(5. Discussion) is still too much to read,  I think it must be retrenched. 

Author Response

The discussion has been reduced as required, please see the attachment with track changes.

This manuscript is a resubmission of an earlier submission. The following is a list of the peer review reports and author responses from that submission.